# Comparative Genomics Reveals Species-Specific Genes and Symbiotic Adaptations in *Tricholoma matsutake*

**DOI:** 10.3390/jof10110746

**Published:** 2024-10-29

**Authors:** Jea Hyeoung Kim, Eun-Kyung Bae, Yoeguang Hue, Byungheon Choi, Min-Jeong Kang, Eung-Jun Park, Ki-Tae Kim

**Affiliations:** 1Department of Plant Medicine, Sunchon National University, Suncheon 57922, Republic of Korea; class_wer@naver.com (J.H.K.); hope9901@naver.com (Y.H.); 2Forest Microbiology and Application Division, Forest Bioresources Department, National Institute of Forest Science, Suwon 16631, Republic of Korea; baeek@korea.kr (E.-K.B.); kangmj1020@korea.kr (M.-J.K.); 3Department of Multimedia Engineering, Sunchon National University, Suncheon 57922, Republic of Korea; caqudgjs@naver.com; 4Department of Agricultural Life Science, Sunchon National University, Suncheon 57922, Republic of Korea

**Keywords:** tricholomataceae, *Tricholoma matsutake*, genomics, ectomycorrhizae, evolution

## Abstract

*Tricholoma matsutake*, a highly valued ectomycorrhizal fungus, requires a symbiotic relationship with pine trees for growth, complicating its cultivation. This study presents a comprehensive comparative genomic analysis of *Tricholoma* species, with a focus on *T. matsutake*. Genomic data from 19 assemblies representing 13 species were analyzed to identify genus-, species-, and strain-specific genes, revealing significant evolutionary adaptations. Notably, *T. matsutake* exhibits a higher proportion of repetitive elements compared to other species, with retrotransposons like LTR Gypsy dominating its genome. Phylogenomic analyses showed that *T. matsutake* forms a monophyletic group closely related to *T. bakamatsutake*. Gene family expansion and contraction analyses highlighted the unique evolutionary pressures on *T. matsutake*, particularly the loss of tryptophan-related metabolic pathways and the gain of genes related to iron ion homeostasis, which may be crucial for its adaptation to nutrient-limited environments. Additionally, the reduction in secreted proteins and carbohydrate-active enzymes reflects the host-dependent lifestyle of *T. matsutake* and related species. These findings enhance our understanding of the genetic and evolutionary mechanisms underlying the complex symbiotic relationships of *T. matsutake*, offering potential avenues for optimizing its cultivation and commercial value.

## 1. Introduction

Species in the genus *Tricholoma* are mushroom fungi belonging to the family Tricholomataceae and the order Agaricales in the phylum Basidiomycota. They are found on every continent across the globe, including Africa, America, Asia, Europe, and Oceania [1,2,3,4]. All *Tricholoma* species are known to be ectomycorrhizal (ECM) fungi, forming symbiotic relationships with various broad-leaved or coniferous trees [5]. A recent phylogenetic study based on 50 multi-loci indicated that the genus *Tricholoma* is divided into at least four subgenera and fourteen sections [6]. Furthermore, with the advancement of next-generation sequencing (NGS) technology, the genomes of numerous *Tricholoma* species have been sequenced and reported (Table 1). However, comprehensive comparative genomic analysis within this genus has not yet been conducted.

Among the species in the genus *Tricholoma*, *T. matsutake*, which has the highest commercial value, necessitates a symbiotic relationship with its host (mainly *Pinus*, but also *Abies*, *Tsuga*, *Picea*, *Quercus*, *and Castanopsis*) for growth, yet its slow growth rate significantly complicates cultivation [7,8]. To address these challenges, various ecological, physiological, and genetic studies have been conducted. Research at the genomic level is particularly important, as comparing the genome of *T. matsutake* with those of closely related species can reveal more specific genetic differences. This comparative genomics approach is instrumental in identifying species-specific genes that have emerged from the co-evolution process between *T. matsutake* and its hosts. For instance, specific genes crucial for interactions with *Pinus* can be identified through gene variations that have adapted during the co-evolution process.

According to the latest telomere-to-telomere assembly, *T. matsutake* strain Sample A has been found to possess 13 chromosomes [9]. However, the number of annotated proteins varied significantly for different studies, ranging from 15,305 to 23,170 [9,10,11,12,13]. This discrepancy is due to different annotation pipelines and differences in gene structural annotation before and after repeat annotation. In this study, a comprehensive comparative analysis of the *T. matsutake* genome was performed using the same annotation method, identifying genus-, species-, and strain-specific genes within the *Tricholoma* genus. Comparative genomic studies with closely related species provide crucial insights for optimizing the growth conditions and improving the cultivation success rate of *T. matsutake*. Ongoing research based on these findings can maximize the commercial value of *T. matsutake* and ensure a stable supply.

## 2. Materials and Methods

### 2.1. Dataset Acquisition and Quality Assessment

To conduct a comparative genomic analysis of species within the genus *Tricholoma*, a search for “Tricholomataceae (Taxonomy ID: 5351)” was performed on NCBI, resulting in the download of 19 assemblies from 13 species (*Cystodermella granulosa*, *Flagelloscypha* sp., *Macrocystidia cucumis*, *Rugosomyces carneus*, *Tricholoma bakamatsutake*, *Tricholoma flavovirens*, *Tricholoma furcatifolium*, *Tricholoma hemisulphureum*, *T. matsutake*, *Tricholoma saponaceum*, *Tricholoma terreum*, and two *Tricholoma* species) (Table 1). The assembly quality was analyzed using BUSCO v5.3.2 and Quast v5.2.0 [14,15].

### 2.2. Structural Annotation of Repetitive Elements, Genes, and RNAs

For a comprehensive comparative genomic analysis, protein-coding genes, ribosomal RNAs (rRNAs), and transfer RNAs (tRNAs) were predicted following the de novo repeat annotation. The repetitive sequences were predicted using the RepeatModeler v2.0.5 and RepeatMasker v4.1.6 pipeline incorporated with LTR_retreiver v2.9.9, RepeatScout v1.0.6, and RECON v1.08 (-LTRStruct mode) [16,17,18,19,20]. The structural annotation of genes was performed with the repeat-masked genomes using BRAKER v3.0.7 based on the 330,798 Tricholomatineae (Taxonomy ID: 2982303) proteins retrieved from the NR database of NCBI and the fungal ab initio method [21]. Augustus v3.5.0 and GeneMark v3.68 were included for BRAKER3 annotation [22,23]. The rRNAs were predicted using Barrnap v0.9 [24], the tRNAs were predicted using tRNAscan-SE v2.0.12 [25], and the eukaryote model was used for both programs.

### 2.3. Synteny Analysis for T. matsutake Genomes

The genome mapping of *T. matsutake* strains to the reference strain Sample A was performed using a reference-based scaffolding tool RagTag v2.1.0 [26]. The synteny among the *T. matsutake* strains and between *T. bakamatsutake* was analyzed using the nucleotide sequences comparison tool nucmer in MUMmer v4.0.0rc1 package [27].

### 2.4. Functional Annotation of the Genes and Enrichment Tests

The functional annotation of genes was performed using DIAMOND alignments against the NR database, InterProScan, and Gene Ontology (GO), and the gene function enrichment analyses were performed by Fisher’s exact test incorporated in OmicsBox v3.1.11. An adjusted *p*-value of <0.05 was used for the cutoff for the functional enrichment test, but a *p*-value of <0.05 was used instead when no enriched GO IDs were found. Unless otherwise specified, the functional enrichment analysis used *T. matsutake* NIFoS 2001 as the reference. Additionally, KofamScan v1.3.0 was used for KEGG annotations [28]. 

SignalP v6.0 with eukaryote mode, dbCAN3 v4.1.4, and EffectorP v3.0 default options were used for the prediction of secreted proteins, carbohydrate-active enzymes (CAZymes), and putative effector proteins, respectively [29,30,31]. For the secondary metabolite biosynthesis gene clusters (SMBGCs), antiSMASH v7.0.0 fungal version with all features and relaxed strictness was implemented [32].

### 2.5. Phylogenomics, Orthology, and Gene Family Expansion/Contraction Analyses

The annotated proteins from the 19 strains were subjected to OrthoFinder v2.5.4 to reveal their orthology and phylogenomics relationship [33]. For gene family expansion and contraction analysis, the output from ortholog analysis and the species tree was subject to CAFE v5.1.0 [34]. The additive species tree was calibrated to an ultrametric tree using treePL v1.0 with a divergence time of 32 MYA between *T. matsutake* and *T. flavovirens* as reference [35]. The divergence time was derived from TimeTree 5 [36].

### 2.6. Idenfication of a Trytophan Metabolism Pathway Gene by PCR Amplification

*T. matsutake* and *T. bakamatsutake* strains (as shown in Appendix A) were cultured on potato dextrose agar (PDA) medium at 25 °C for 2 months. For DNA extraction, 100 mg of mycelium was frozen in liquid nitrogen and ground to a fine powder using a TissuLyser II (Qiagen, Hilden, Germany). Genomic DNA was then extracted from lyophilized mycelia using the Allprep Fungal DNA/RNA/Protein kit (Qiagen) following the manufacturer’s protocol. Primers were designed using Primer 3 web version 4.1.0 (primer3.ut.ee) and synthesized by Macrogen Co. Ltd. (Seoul, Republic of Korea). PCR amplification of the 3-hydroxyanthranilate 3,4-dioxygenase (3HAO) gene was performed using specific primers (Appendix A) and AccuPower Taq PCR PreMix (Bionner, Daejeon, Republic of Korea). The PCR conditions were as follows: initial denaturation at 95°C for 3 min, followed by 35 cycles of denaturation at 95 °C for 30 s, annealing at 59 °C for 30 s, and extension at 72 °C for 90 s, with a final extension at 72 °C for 5 min. PCR products were analyzed on a 1.0% agarose gel in 1% TAE buffer, stained with EcoDye DNA Staining Solution (SolGent, Daejeon, Republic of Korea), and run at 100 V for 20 min.

## 3. Results

### 3.1. Genome Quality and Metadata of the Available Agaricales Strains

*T. matsutake* strains were isolated from China, Korea, and Japan, while *T. bakamatsutake* strains were isolated from China and Japan (Table 1). Additionally, *Tricholoma MG* strains were isolated from China for mushroom genome analysis study [13]. Among the *Tricholoma* species, only the *T. hemisulphureum* strain was isolated outside Asia, specifically from the United Kingdom [37]. The species assembled at the chromosome level include *T. matsutake* sample A (13 chromosomes), *T. bakamatsutake* Sf-Tf05 (14 chromosomes) [38], and *T. hemisulphureum* (12 chromosomes) (Table 1). According to BUSCO analysis, essential genes at the Agaricales level were well preserved across all strains of *T. matsutake* and *T. bakamatsutake*, as well as *T. flavovirens* and *T. hemisulphureum*, regardless of the number of scaffolds (Appendix A). Although the BUSCO scores for *Tricholoma* outgroup species at the Agaricales level were relatively insufficient, all were used to predict *Tricholoma* genus-specific genes.

**Table 1 jof-10-00746-t001:** The genome statistics of *Tricholoma* and outgroup strains used in this study.

Species	Strain	Isolated Location	No. of Scaffolds	Assembly Size (bp)	N50 (bp)	L50	GC (%)	Repeat Size (bp)	Repeat Content (%)	Reference (GenBank Accession)
*Tricholoma matsutake*	NIFoS 2001	Korea	80	162,632,606	5,103,859	10	45.50	116,305,332	71.51	Kang et al., 2023 [12] (GCA_026213095.1)
*Tricholoma matsutake*	KMCC04578	Korea	5255	189,030,256	93,406	589	45.25	99,821,686	69.22	Min et al., 2020 [10] (GCA_002939025.2)
*Tricholoma matsutake*	MG52	China	58,276	155,036,874	9294	3971	45.35	108,421,248	71.23	Li et al., 2018 [13] (GCA_003314635.1)
*Tricholoma matsutake*	NBRC 30605	Japan	88,884	131,718,925	2909	9526	45.46	88,187,244	67.19	Unpublished (GCA_001652835.1)
*Tricholoma matsutake*	945	NA	2545	175,758,422	320,875	167	45.40	104,910,411	69.01	Miyauchi et al., 2020 [11] (GCA_014904895.1)
*Tricholoma matsutake*	Sample A	Japan	13	161,040,721	12,376,747	6	45.50	115,538,238	71.74	Kurokochi et al., 2023 [9] (GCA_026075535.2)
*Tricholoma bakamatsutake*	SF-Tf05	Japan	14	142,190,777	12,427,164	6	43.92	108,654,712	76.41	Ichida et al., 2023 [38] (GCA_029959215.1)
*Tricholoma bakamatsutake*	MG51	China	51,308	140,676,125	9944	3260	43.93	106,664,015	77.08	Li et al., 2018 [13] (GCA_003313665.1)
*Tricholoma flavovirens*	MG32	China	49,305	119,315,121	7637	3611	45.00	80,893,955	68.90	Li et al., 2018 [13] (GCA_003313805.1)
*Tricholoma sinoportentosum* *	MG77	China	53,256	118,334,115	5062	6051	47.09	65,740,839	57.47	Li et al., 2018 [13] (GCA_003314665.1)
*Tricholoma saponaceum*	MG146	China	14,500	55,708,857	11,929	1228	47.00	10,707,825	19.35	Li et al., 2018 [13] (GCA_003313625.1)
*Tricholoma albobrunneum* **	MG99	China	18,072	59,536,732	9454	1540	46.76	16,205,717	27.82	Li et al., 2018 [13] (GCA_003521275.1)
*Tricholoma hemisulphureum*	KDTOL00252	UK	26	128,976,826	10,683,071	5	47.13	88,317,381	68.48	Unpublished (GCA_964034925.1)
*Tricholoma terreum*	MG45	China	25,404	84,437,827	9877	2259	47.87	38,467,970	46.28	Li et al., 2018 [13] (GCA_003316345.1)
*Praearthromyces corneri* ***	D33	Malaysia	17,094	49,805,319	10,870	876	47.44	11,299,287	22.73	Unpublished (GCA_018854895.1)
*Rugosomyces carneus*	D47	Netherlands	39,733	58,963,700	3469	2576	47.02	15,608,583	26.69	Unpublished (GCA_018221805.1)
*Cystodermella granulosa*	DBG-29008	USA	23,419	35,367,994	2089	4024	45.65	4,136,583	11.71	Kraisitudomsook et al., 2024 [37] (GCA_036937035.1)
*Flagelloscypha* sp.	PMI 526	NA	397	72,728,845	1,132,480	18	46.84	9,103,480	12.52	Unpublished (GCA_021399455.1)
*Macrocystidia cucumis*	KM177596	UK	204,508	46,746,859	15,511	599	48.76	4,791,427	10.33	Unpublished (GCA_001179725.1)

* Originally *Tricholoma* sp. MG77; ** *Tricholoma* sp. MG99; and *** *Tricholoma furcatifolium*.

### 3.2. Evolutionary Relationship of the Agaricales Strains

The evolutionary relationships of Agaricales species were established using 1056 ortholog clusters shared among 19 strains. The species tree indicates that the genus *Tricholoma*, excluding *T. furcatifolium*, is monophyletic (Figure 1A). Previous DNA analyses suggested that *T. furcatifolium* is not associated with the *Tricholoma* genus and proposed its classification into a new genus, *Praearthromyces* [39]. Additionally, a BLAST search of the *TEF1A* sequence of *T. furcatifolium*, presumed to be *Praearthromyces*, in the NR database showed 100% coverage and identity with *Praearthromyces corneri*. This indicates that *T. furcatifolium* in the NCBI genome dataset is *P. corneri* (Appendix A). *Tricholoma* sp. MG77 appears closely related to *T. flavovirens*; however, the relatively long branch length and differences in GC content suggest species-level divergence. Genome-based ITS analysis revealed that the MG77 strain is identified as *T. sinoportentosum*, which belongs to the same section *Tricholoma* as *T. flavovirens*, while the MG99 strain is identified as *T. albobrunneum*, belonging to the section *Genuina* (Appendix A). Henceforth, the newly identified name will be used in the manuscript.

### 3.3. Evolutionary Relationship of the Strains in Section Matsutake

*T. matsutake* strains formed a monophyletic group, appearing as a sister clade to *T. bakamatsutake* strains. Among the *T. matsutake* strains, two Korean strains, NIFoS 2001 and KMCC04578, formed a clade, with the Chinese strain MG2 as their outgroup and the Japanese strain NBRC 30605 and sample A each appearing as additional outgroups (Figure 1B). Although the origin of strain 945 is unclear [11], phylogenomic analysis suggests a high likelihood that it was collected from Japan. In addition, the genomes of *T. matsutake* strains were mapped to Sample A with 161 Mbp size chromosome-level assembly as reference (Figure 1B). The genomes of two Korean strains and strain 945 were almost fully mapped to the reference, but the strains KMCC04578 and 945 had additional genome sequences. The genomes of Chinese strain MG52 and Japanese strain NBRC 30605 were not fully mapped to the reference, but they had their own unique genome sequences. Overall, the syntenic relationship was preserved within *T. matsutake* strains and was partially conserved with *T. bakamatsutake*, which belongs to the same section *Matsutake* (Appendix A). Additionally, the synteny was not conserved with *Tricholoma* species outside this section.

### 3.4. Genome Statics of the Selected Agaricales Strains

The genome size of Agaricales species shows a broad range from 35.3 Mbp in *C. granulosa* to 189.0 Mbp in *T. matsutake* KMCC04578 (Figure 1A, Table 1). Within the *Tricholoma* genus, all species except *T. terreum*, *T. albobrunneum* (MG99), and *T. saponaceum* possess genomes larger than 100 Mbp. Notably, species within the *Matsutake* section, including *T. matsutake* and *T. bakamatsutake*, have larger genomes than other *Tricholoma* species. Compared to the outgroup, the relatively large genome sizes of *Tricholoma* species can be attributed to an increased content of repetitive DNA.

The GC content of Agaricales varies across species (Figure 1A, Table 1). Nevertheless, strains within the same species exhibit nearly identical GC content. *T. matsutake* strains isolated from different regions have a GC content ranging from 45.3% to 45.5%, while *T. bakamatsutake* strains show a GC content of 43.9%. Regarding tRNA, excluding *M. cucumis*, which had poor assembly, the number of tRNA genes ranged from a minimum of 284 to a maximum of 577, and rRNA genes ranged from 0 to a maximum of 45 (Table 2). The variability in the number of tRNA and rRNA genes is attributed to these genes being recognized as known repeats [40], which can cause repetitive sequences to be inadequately represented depending on the sequencing methods used.

After repeat masking and re-annotation, the number of protein-coding genes in *Tricholoma* species ranged from 12,460 in *T. hemisulphureum* to 17,939 in *T. sinoportentosum* (MG77) (Figure 1A, Table 2). Previously, *T. matsutake* strains showed a wide range in gene counts; however, on average, they were found to have 13,809 genes coding for 16,221 proteins. Of these, only 65.6% have associated Gene Ontology terms, with the remaining genes presumed to encode proteins of unknown function.

### 3.5. Repeat Content within the Selected Strains

Analysis of the repeat content in selected species revealed that, except for *T. saponaceum* and *T. albobrunneum* (MG99), *Tricholoma* species possess more repeat elements compared to outgroup species (Figure 2A). The average repeat content in *Tricholoma* species was approximately 60%. Excluding the unclassified repeats, the most abundant repeat elements in *Tricholoma* genomes are retrotransposons, including LTRs and LINEs, with LTR Gypsy being the most prevalent and constituting a high proportion. DNA transposons are evenly distributed among *Tricholoma* species, except for the two species with smaller repeat content mentioned above, while LINEs appear increased only in *T. matsutake* strains. Additionally, *T. bakamatsutake* has a higher proportion of rolling circle elements compared to other *Tricholoma* species.

The distribution of repeat sequences within genomes according to the Kimura substitution level indicates the relative age of repeat element generation [41]. *Flagelloscypha* sp. has lower repeat content compared to *Tricholoma* species, but a high proportion of its repeats were generated long ago, indicating a relatively stable genomic environment (Figure 2B). In contrast, *Tricholoma* species show high proportions of repeat sequences at low Kimura substitution levels, suggesting a recent surge in repeat element activity in each species.

### 3.6. The Pan and Core Orthogroups of Selected Strains

The ortholog analysis of 325,167 proteins in 19 strains revealed 34,538 orthogroups, including 22,708 unassigned proteins (singletons) (Table 3). Among them, 11,788 *Tricholoma* genus-specific orthogroups were identified, with 10,027 orthogroups sharing with the outgroup (Figure 3A). The number of *Tricholoma* genus-specific orthogroups without singletons was 6359 and was divided into six sections. The division identified 137 *Tricholoma* core orthogroups, and their putative functions were fungal-type cell wall, pentose phosphate shunt, gamma-aminobutyric acid catabolic process, glutamate decarboxylation to succinate, and thiamine triphosphatase activity (Figure 3B). These functions are ubiquitous in fungi, and gamma-aminobutyric acid and succinate have been reported to be produced by *T. matsutake* [42].

The number of *Matsutake* section-specific orthogroups was 2803, and 120 orthogroups shared between *T. bakamatsutake* and *T. matsutake* were identified (Figure 3A). The proteins of the shared orthogroup are primarily active in the extracellular region and are involved in functions such as flavin adenine dinucleotide binding, pyranose dehydrogenase activity, chitin-binding and the catabolic process, choline dehydrogenase activity, glycine betaine biosynthesis, and the sphingoid catabolic process (Figure 3B). Molecules such as chitin, sphingoid, choline, and betaine related to the functions are major fungal components found in *T. matsutake* extracts [42].

Lastly, the 2245 species-specific orthogroups of *T. matsutake* were divided among six strains, yielding region- and strain-specific orthogroups and 195 *T. matsutake* species-specific core orthogroups (Figure 3A). The major functions of the core orthogroups included the negative regulation of intracellular signal transduction and TORC1 signaling, the D-galactonate catabolic process, glycogen (starch) synthase activity, the GATOR1 complex, the pyridoxine biosynthetic process, and quercetin 3-O-glucosyltransferase activity (Figure 3B). These functions highlight specialized metabolic and regulatory strategies of *T. matsutake* that are essential for its ecological niche and symbiotic lifestyle.

### 3.7. The Gene Family Expansion and Contraction in Tricholoma Evolution

The CAFE analysis revealed the numbers of expanded and contracted gene families in the evolution of *Tricholoma* and outgroup strains (Figure 4A). During the divergence of the *Tricholoma* genus, 457 gene families expanded and 1265 gene families contracted. At the divergence of the *Matsutake* section, 383 gene families expanded and 1056 contracted, while, at the divergence of *T. matsutake*, 592 gene families expanded and 239 contracted.

To understand the functional characteristics of the *Tricholoma* genome, the functions of gene families that expanded and contracted at the divergence points of the *Tricholoma* genus, the *Matsutake* section, and *T. matsutake* were analyzed (Figure 4B). For the expanded gene families, *T. matsutake* NIFoS 2001 was used as a reference to identify enriched functions. Gene families commonly expanded at all divergence points were associated with DNA metabolism and repair as well as medium-chain fatty acid metabolism, while each divergence point exhibited unique functions specific to that time (Figure 4B). In contrast, for the contracted gene families, the functions of gene families completely lost in the *Tricholoma* genus, *Matsutake* section, and *T. matsutake* were identified using *P. corneri*, *T. flavovirens*, and *T. bakamatsutake* SF-Tf05 as outgroups, respectively. No common functions were found among the contracted gene families (Figure 4A).

The functions of gene families expanded in the *Tricholoma* genus were primarily associated with amide, peptide, glycerol, and water transport, as well as the nuclear envelope and RNA polymerase. The functions of completely deleted gene families were related to the catabolic processes of plant components such as cellulose, pectin, and glycerolipids, along with various enzyme activities. In the *Matsutake* section, the functions of expanded and lost gene families were diverse. The main functions of the expanded gene families were related to transmembrane transport, hydrolase activity, and fungal-type vacuoles, while the lost gene families were associated with various degradative enzymes such as oxygenase, peptidase, and glucosidase, as well as metabolic processes. In *T. matsutake*, gene families related to iron ion binding and homeostasis, small RNA regulation, and the cell cycle were expanded. To investigate the genes related to iron binding, a major portion of the expanded gene families in *T. matsutake*, proteins involved in the four main iron uptake mechanisms in fungi (Fet4p, Fre1p, Fet3p, Ftr1p, Rbt5, and Sit1p) were identified through a BLAST search (E-value < 0.05) in both *T. matsutake* and *T. bakamatsutake* (Figure 5) [43]. No homologs of Fet4p, responsible for direct ferrous iron permeation, were found at an E-value below 0.05 (though they were detected at an E-value above 1). This corresponds to the gene families that are contracted in *T. matsutake* (Figure 4B). Homologs of Fre1p and Fet3p, associated with iron ion binding, were more abundant in *T. matsutake* compared to *T. bakamatsutake*. However, iron permease Ftr1p, heme receptor Rbt5, and siderophore uptake transporter Sit1p were more frequently identified in *T. bakamatsutake*. 

On the other hand, the lost gene families were related to the quinolinate biosynthetic process, anthranilate metabolic process, and, consequently, the biosynthesis of NAD, all of which are associated with tryptophan metabolism (Figure 4B). KEGG analysis of the tryptophan metabolism pathway revealed that the pathway from tryptophan to quinolinate and NAD+ was preserved in all *T. bakamatsutake* strains, while, in *T. matsutake* strains, the absence of 3-hydroxyanthranilate 3,4-dioxygenase (3HAO; EC:1.13.11.6) resulted in an incomplete pathway (Figure 6, Appendix A). Two homologous copies of 3HAO were found in *T. bakamatsutake*, while none were present in *T. matsutake*. However, BLAST results for 3HAO in *T. bakamatsutake* show the existence of a homolog with low conservation in *T. matsutake* (Figure 6). Additional PCR amplification of the *3HAO* gene from *T. bakamatsutake* revealed that the gene is present in *T. bakamatsutake* strains but absent in *T. matsutake* strains (Appendix A). Although tryptophan has been found in *T. matsutake* extracts [44], the absence of 3HAO suggests that *T. matsutake* may depend on its host to supply intermediates, particularly in the steps leading to 3-hydroxyanthranilate and quinolinate.

### 3.8. CAZyme, Secreted Protein, Effector, and Secondary Metabolite Gene Cluster Content in Selected Strains

The analysis of secreted proteins, CAZymes, and putative effectors in *Tricholoma* species revealed that, consistent with the gene family loss observed at the divergence of the *Tricholoma* genus, *Tricholoma* species had fewer predicted proteins compared to the outgroup (Figure 7). While the outgroup species possessed more than 1000 secreted proteins, *Tricholoma* species had approximately 500 to fewer than 1000 proteins. Notably, *T. bakamatsutake* had the smallest number of secreted proteins, ranging from about 532 to 555. The total number of CAZymes and secreted CAZymes in *Tricholoma* species ranged from 634 to 801 and 143 to 190, respectively, which was significantly lower than in the outgroup. The number of secreted effectors in *Tricholoma* species also ranged from 90 to 229, which was markedly lower than in the outgroup.

In the case of SMBGCs, most selected species here were found to possess one or more type 1 PKS and NI-siderophore clusters. The fungal-RiPP-like cluster was absent in the *Matsutake* section and presented only in the outgroup, while the indole cluster was found exclusively in the *Flagelloscypha* sp. and *M. cucumis* clade. On the other hand, NRPS-like and terpene clusters were widely present within *Tricholoma* and outgroup species. Notably, the (+)-δ-cadinol sesquiterpene synthase, which is found in many conifers and other fungi, was detected in all strains except for the *T. matsutake* Korean strains, Sample A, *T. bakamatsutake* SF-Tf05, and *C. granulosa* [45,46]. This may serve as evidence of horizontal gene transfer with hosts, and the diversity of SMBGCs across *Tricholoma* species suggests their potential to produce various secondary metabolites.

## 4. Discussion

The comparative genomic analysis of *Tricholoma* species, mainly focusing on *T. matsutake*, reveals critical insights into the evolutionary mechanisms shaping the genomic architecture and functional specialization of these fungi. The size of non-repetitive DNA, excluding repeats, in *Tricholoma* genomes averaged around 48 Mbp, similar to the average genome size of outgroup species and other Basidiomycetes [47]. The similarity in the size of non-repetitive DNA between *Tricholoma* and outgroup species suggests that the expansion of *Tricholoma* genomes is attributed to the proliferation of repetitive sequences rather than an increase in non-repetitive DNA. Especially the relatively large genome sizes in species like *T. matsutake* and *T. bakamatsutake* can be primarily attributed to the expansion of repetitive elements, particularly LTR Gypsy retrotransposons (Figure 2A). Previous studies have indicated that merely LTR elements constitute the most significant portion of the *T. matsutake* genome [10,11]. The recent activity of these elements, indicated by their low Kimura substitution levels, points to ongoing genomic evolution. In addition, the increased content of LINEs in *T. matsutake* strains and the higher proportion of rolling circle elements in *T. bakamatsutake* highlight specific evolutionary adaptions in these species (Figure 2B). For instance, the symbiotic relationship between *T. matsutake* and its hosts, which thrive in specific forest ecosystems [48], may have driven the selection of these genetic changes, promoting resilience to external stressors like soil nutrient depletion. These ongoing genomic changes not only reflect evolutionary adaptation but also may enhance the functional specialization of these fungi in their unique ecological niches. 

The phylogenomic analyses revealed that the genus *Tricholoma* forms a monophyletic group, supporting previous taxonomic classifications (Figure 1A) [39]. The divergence of *T. matsutake* from its close relatives, such as *T. bakamatsutake*, is particularly intriguing as it suggests that specific evolutionary pressures have shaped the genome of *T. matsutake* differently from other members of the genus. The presence of region-specific clades within *T. matsutake* strains underscores the impact of geographic isolation on genetic divergence (Figure 1B). These findings provide a foundation for further exploration into how environmental factors and host interactions drive speciation and genetic diversity within this genus.

In this study, GO annotation was used to explore the functional enrichment of genes in *T. matsutake* and related species. Despite only 65.6% of the proteins in *T. matsutake* being assigned GO terms, significant functions were identified in core orthogroups and orthogroups that underwent expansion or contraction through enrichment tests (Figure 3 and Figure 4). Although the functions of most proteins within these orthogroups remain unknown, the proteins that have been assigned GO terms reflect the biological characteristics of *T. matsutake* and its related species at the genomic level. However, the analysis is limited by the incomplete annotation of proteins, as many remain functionally uncharacterized, which may restrict a comprehensive understanding of the full biological roles of these orthogroups.

The identification of core orthogroups within the section *Matsutake*, particularly those involved in extracellular functions and metabolic processes, reflects the specialized lifestyle of these fungi (Figure 3B). Functions such as chitin-binding, terpene synthesis, and sphingoid catabolic processes are crucial for interacting with their environment, breaking down organic matter, and synthesizing bioactive compounds. These processes support nutrient acquisition and cellular metabolism, enabling the fungi to adapt to various environmental conditions and maintain their symbiotic relationship with host plants. Additionally, the involvement of mitochondrial-nucleus signaling pathways and protein tagging mechanisms indicates the importance of cellular regulation and stress adaptation in these fungi’s survival and ecological roles.

The identification of *T. matsutake*-specific orthogroups related to TORC1 signaling, the D-galactonate catabolic process, starch synthase activity, the GATOR1 complex, the pyridoxine biosynthetic process, and quercetin 3-O-glucosyltransferase activity provides valuable insights into how this fungus may have adapted to modulate its growth and metabolism in response to the nutrient-limited environments that it typically inhabits (Figure 3B). These pathways suggest that *T. matsutake* has evolved specialized mechanisms to optimize nutrient acquisition and utilization under challenging ecological conditions. Quercetin, which has been identified in *T. matsutake* extracts [42], alongside essential vitamins such as thiamin, nicotinic acid, and pyridoxine, as well as starch, has been shown to enhance mycelial growth and promote amylase production, a critical factor for fruit body formation [7,49,50,51]. Additionally, TORC1 is a crucial regulator of cell growth and development in fungi, controlling nutrient absorption, metabolism, protein synthesis and degradation, morphology, and cytotoxicity, and the GATOR complex serves as an inhibitor of TORC1 [52]. The negative regulation of TORC1 likely influences the slow growth of Matsutake mushrooms. This negative regulation by the GATOR complex could lead to reduced TORC1 activity, thereby slowing down cellular processes and overall growth rates. This modulation allows the fungi to adapt to nutrient-limited environments, which is essential for their survival and symbiotic relationship with their host plants.

The gene family expansion and contraction analysis provides deeper insights into the adaptive strategies employed by *T. matsutake* throughout its evolution. The expansion of gene families related to iron ion homeostasis and small RNA regulation suggests that the fungus has developed sophisticated mechanisms to manage nutrient availability and regulate gene expression, which are essential for survival in its specific ecological niche (Figure 4B). Several studies have reported that iron ions play a significant role in promoting *T. matsutake* growth, with evidence showing that the soil that it inhibits is generally nutrient-poor but rich in iron ions [7,53,54,55]. In these environments, the ability to efficiently acquire, store, and regulate iron offers a significant competitive advantage. Enhanced iron uptake mechanisms, potentially through more efficient transporters, likely enable *T. matsutake* to outcompete other organisms for this essential nutrient (Figure 5). Moreover, the expansion of iron-related gene families may provide precise control over iron-dependent cellular processes, optimizing growth and metabolism in iron-rich environments.

The gene loss observed in *T. matsutake* related to tryptophan metabolism, particularly the absence of 3HAO, suggests a significant evolutionary shift toward a symbiotic lifestyle (Figure 4B and Figure 6). This gene loss may result in a metabolic dependency on its host or environment for intermediates like quinolinate and NAD+, classifying *T. matsutake* as an auxotroph for these compounds. The contraction of these gene families may reflect an evolutionary trade-off, where the fungus has streamlined its metabolism to conserve energy by outsourcing the biosynthesis of certain compounds to its host, enhancing its ecological fitness within a stable symbiotic relationship. However, as these findings are based solely on comparative genome analysis, experimental validation is necessary to confirm the functional implications of the gene loss and metabolic dependency in *T. matsutake*. Future studies should focus on investigating whether the fungus indeed acquires tryptophan intermediates from its host and how this metabolic reliance impacts the dynamics of its symbiotic relationship.

Compared to outgroup species, the reduction in the number of secreted proteins and CAZymes in *T. matsutake* and related species indicates their adaptation to a host-dependent lifestyle (Figure 7). The limited diversity of CAZymes suggests that these fungi have streamlined their enzymatic toolkit to suit the specific needs of their symbiotic relationship, possibly focusing on the breakdown of particular substrates provided by their host plants. These results reflect the host-dependent lifestyle of *Tricholoma* species, a phenomenon also observed in other ectomycorrhizae [56]. The analysis of SMBGCs reveals the potential for *T. matsutake* to produce unique bioactive compounds, which may play roles in host interaction, defense mechanisms, or ecological competition (Figure 7). The absence of certain SMBGCs, like the fungal-RiPP-like cluster in the section *Matsutake*, could indicate a divergence in secondary metabolism pathways that corresponds to the distinct ecological roles that these fungi play.

The comparative genomics analysis of *Tricholoma* species, with a focus on *T. matsutake*, reveals significant evolutionary and functional adaptations within this fungal group. Key findings include a higher proportion of repeat elements, special functions identified in species-specific orthogroups, the expansion and contraction of gene families, and a reduction in secreted proteins and CAZymes compared to outgroup species, reflecting their host-dependent lifestyle. These insights underscore the specialized ecological and metabolic traits of *Tricholoma* species, driven by their unique evolutionary pressures and symbiotic relationships.

## Figures and Tables

**Figure 1 jof-10-00746-f001:**
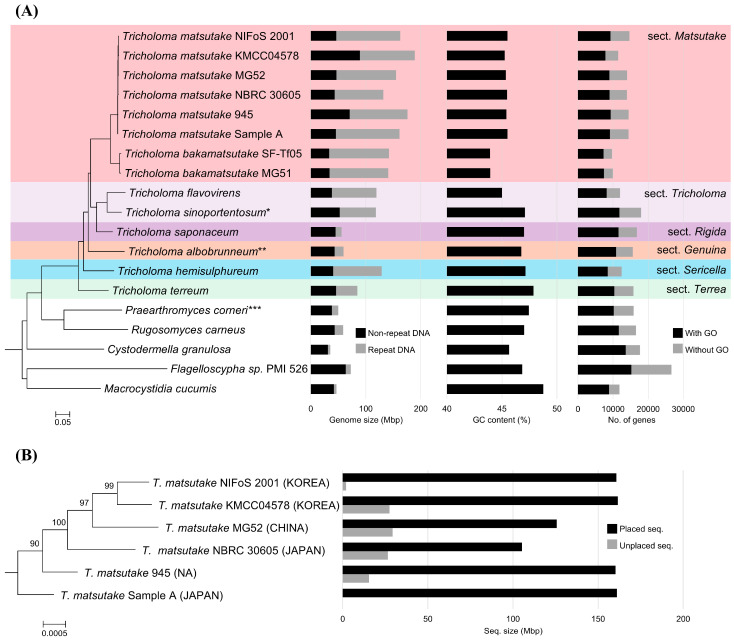
Phylogenomic tree of Agaricales strains along with genome size, GC content, and number of annotated genes. (**A**) The species tree was constructed using concatenated orthologous protein sequences conserved among the selected species/strains, employing the maximum likelihood method. The sections of the *Tricholoma* genus are based on findings from a previous study [6]. *T. sinoportentosum* *, *T. albobrunneum* **, and *P. corneri* *** are originally *Tricholoma* sp. MG77, MG99, and *T. furcatifolium*, respectively. (**B**) An expanded tree focusing on *T. matsutake* strains is presented, showing bootstrap values and the proportion of genomes that align with the reference strain Sample A. The bar graph depicts the size of matched (black) and unmatched (gray) sequences.

**Figure 2 jof-10-00746-f002:**
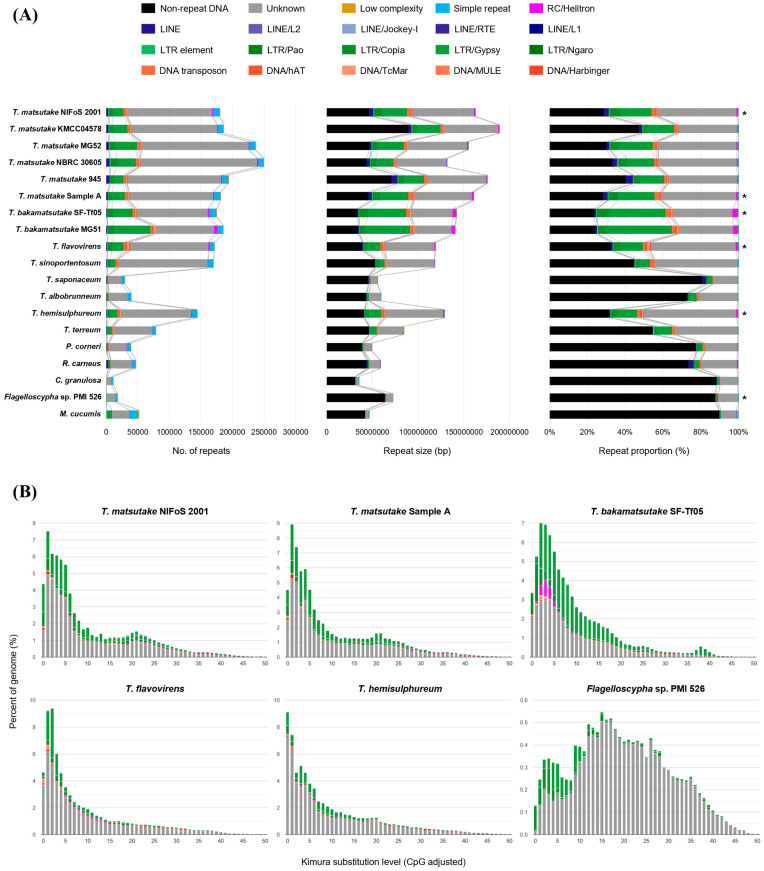
Repeat content in *Tricholoma* and outgroup strains and their evolution. (**A**) The repeat content of *Tricholoma* and outgroup species/strains is shown, categorized by the number of repeats (left), the total occupied size (middle), and the occupied proportion in the genomes (right). (**B**) The asterisks in (**A**) indicate the strains that are further analyzed for Kimura substitution levels. The non-repeatitive DNA is depicted in black, while different types of repeats are color-coded.

**Figure 3 jof-10-00746-f003:**
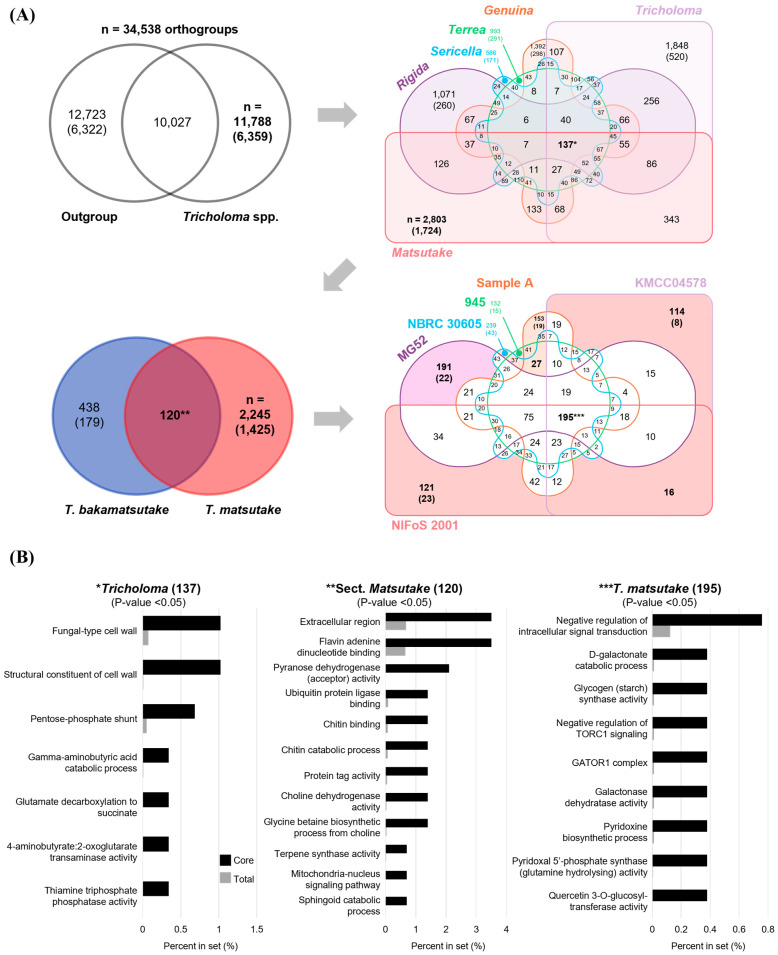
Pan/core genome of *Tricholoma* and outrgroup strains and functional analysis of core orthogroups. (**A**) Venn diagrams illustrate the numbers of orthogroups that are shared among or unique to the evolutionary clades of *Tricholoma* and outgroup strains. Numbers in parentheses indicate orthogroups excluding singletons. Asterisks denote the number of unique core orthogroups in the genus *Tricholoma **, section *Matsutake ***, and *T. matsutake* species *** clades. (**B**) The bar graphs shows the enriched functions within each unique core orthogroup compared to the total proteome (Fisher’s exact test with *p*-value < 0.05). Black and gray bars represent the percentage of proteins in the core orthogroup and the total proteome, respectively.

**Figure 4 jof-10-00746-f004:**
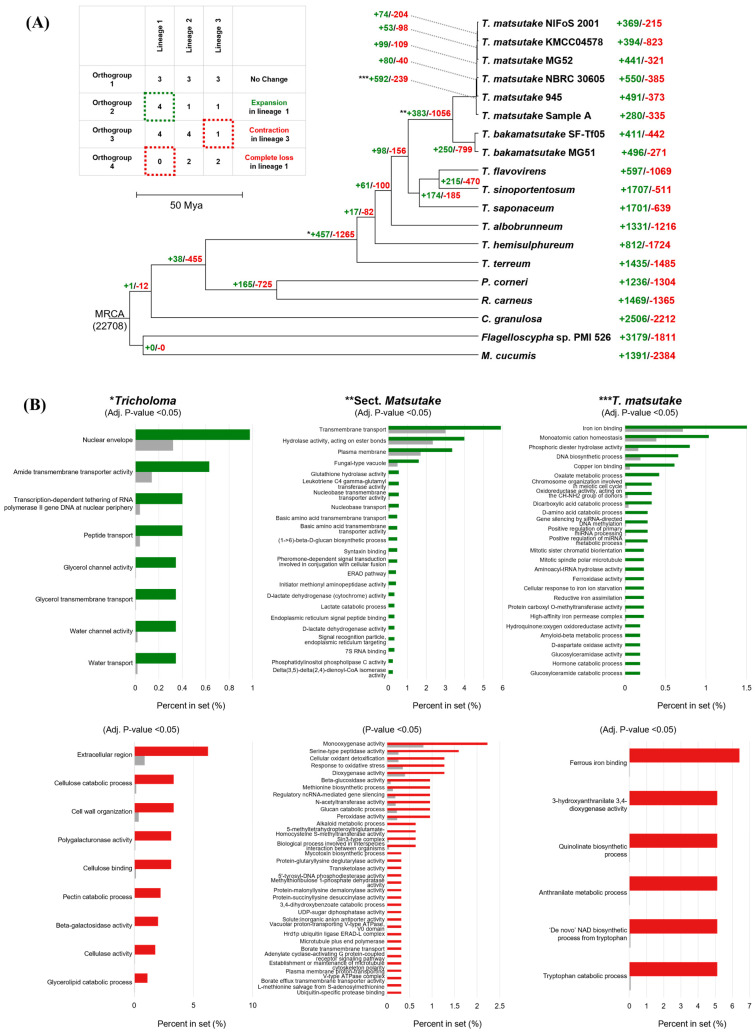
Gene family expansion and contraction in *Tricholoma* and outgroup strains and functional analysis at key evolutionary divergence points. (**A**) The phylogenomic tree depicts the dynamics of gene family expansion and contraction across various evolutionary lineages within selected Agaricales. Numbers in green indicate expanded gene families, while numbers in red indicate contracted gene families. The common ancestor of the genus *Tricholoma* is marked as *, the common ancestor of section *Matsutake* is marked as **, and the common ancestor of *T. matsutake* species is marked as ***. Additionally, a functional analysis of the expanded and lost gene families in these groups was conducted. (**B**) Bar graphs present the functions of gene families that have expanded (green bars) or are completely lost (red bars) during the evolutionary divergence of the *Tricholoma* genus (left), the section *Matsutake* (middle), and the species *T. matsutake* (right). Each bar represents a specific gene function, with the length of the bar corresponding to the relative abundance of that function in the expanded or completely lost gene families compared to the total proteins (gray).

**Figure 5 jof-10-00746-f005:**
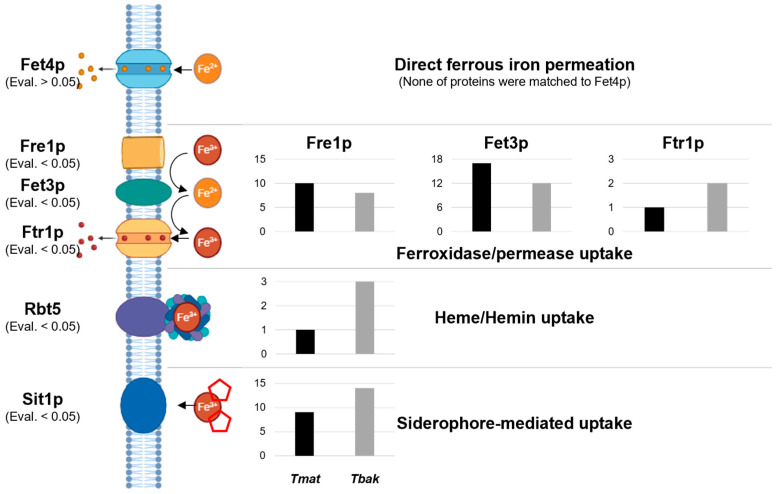
The number of protein homologs involved in the four major iron uptake mechanisms in fungi in *T. matsutake* and *T. bakamatsutake*. Rajapitamahuni et al. investigated the proteins involved in fungal iron uptake mechanisms [43], and the bar graphs display the number of homologs for each protein identified in *T. matsutake* (Tmat, black bars) and *T. bakamatsutake* (Tbak, gray bars) through a BLAST search. E-value cutoffs are shown below the protein names.

**Figure 6 jof-10-00746-f006:**
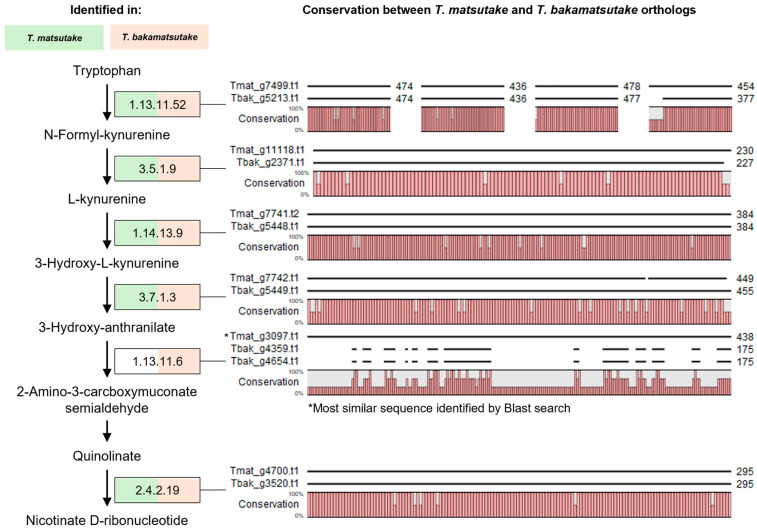
Tryptophan to nicotinate D-ribonucleotide pathway in *T. matsutake* and *T. bakamatsutake*. The pathway from tryptophan to nicotinate D-ribonucleotide in *T. matsutake* and *T. bakamatsutake* is depicted based on the tryptophan, nicotinate, and nicotinamide metabolism pathway from KEGG (Appendix A). Each metabolic step is annotated with its corresponding EC number, and genes present in *T. matsutake* NIFoS 2001 (Tmat) and *T. bakamatsutake* Sample A (Tbak) are highlighted in green and red, respectively. Next to each EC number, ortholog sequence alignments between Tmat and Tbak are shown, along with their sequence conservation. EC1.13.11.52 has four paralogs in both Tmat and Tbak, while EC1.13.11.6 has only two paralogs in Tbak, with none in Tmat. The most similar sequence in Tmat was identified through a BLAST search and aligned.

**Figure 7 jof-10-00746-f007:**
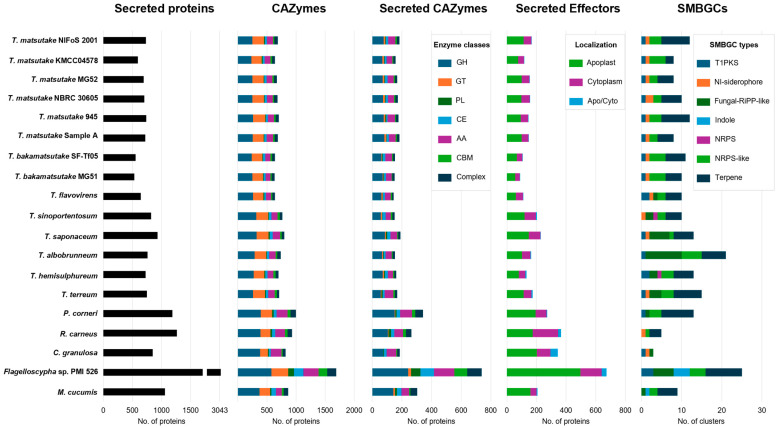
Carbohydrate-active enzymes, secreted proteins, and secondary metabolite biosynthetic gene clusters of the Agaricales strains. The bar graphs provide a comparative analysis of various functional categories across different Agaricales strains. The first panel shows the number of secreted proteins identified in each strain. The second and third panels display the number of CAZymes and secreted CAZymes, categorized into six classes: glycoside hydrolases (GHs), glycosyl transferases (GTs), polysaccharide lyases (PLs), carbohydrate esterases (CEs), auxiliary activities (AAs), and carbohydrate-binding modules (CBMs). The fourth panel shows the distribution of secreted effectors, classified based on their predicted localization: apoplast, cytoplasm, or both (Apo/Cyto). The last panel presents the diversity of SMBGCs across the species, including type I polyketide synthases (T1PKSs), non-ribosomal peptide synthetases (NRPSs), NRPS-like, terpene synthases, NI-siderophores, fungal-RiPP-like clusters, and indole clusters.

**Table 2 jof-10-00746-t002:** The annotation result of *Tricholoma* and outgroup strains used in this study.

Species	Strain	tRNAs	rRNAs	Protein-Coding Genes	Proteins	BLAST (%)	GO (%)	With EC no. (%)	InterPro (%)
*Tricholoma matsutake*	NIFoS 2001	277	25	14,657	17,257	16,410 (95.1)	11,086 (64.2)	3717 (21.5)	10,890 (63.1)
*Tricholoma matsutake*	KMCC04578	275	4	11,449	13,364	12,862 (96.2)	9318 (69.7)	3444 (25.8)	9230 (69.1)
*Tricholoma matsutake*	MG52	300	18	13,992	16,437	15,636 (95.1)	10,739 (65.3)	3774 (23.0)	10,491 (63.8)
*Tricholoma matsutake*	NBRC 30605	304	9	13,931	16,356	15,559 (95.1)	10,780 (65.9)	3893 (23.8)	10,325 (63.1)
*Tricholoma matsutake*	945	281	3	14,442	17,048	16,292 (95.6)	11,138 (65.3)	3818 (22.4)	11,028 (64.7)
*Tricholoma matsutake*	Sample A	275	11	14,385	16,863	16,003 (94.9)	10,820 (64.2)	3639 (21.6)	10,774 (63.9)
*Tricholoma bakamatsutake*	SF-Tf05	287	31	9697	11,071	10,646 (96.2)	8464 (76.5)	3145 (28.4)	8486 (76.7)
*Tricholoma bakamatsutake*	MG51	283	45	9931	11,635	11,254 (96.7)	8842 (76.0)	3343 (28.7)	8758 (75.3)
*Tricholoma flavovirens*	MG32	304	9	11,972	13,643	12,747 (93.4)	9523 (69.8)	3424 (25.1)	9481 (69.5)
*Tricholoma sinoportentosum*	MG77	577	13	17,939	20,382	18,748 (92.0)	13,799 (67.7)	5037 (24.7)	12,998 (63.8)
*Tricholoma saponaceum*	MG146	616	1	16,777	19,191	17,956 (93.6)	13,533 (70.5)	4827 (25.2)	13,564 (70.7)
*Tricholoma albobrunneum*	MG99	516	22	15,597	17,587	16,261 (92.5)	12,478 (71.0)	4542 (25.8)	12,271 (69.8)
*Tricholoma hemisulphureum*	KDTOL00252	305	1	12,460	14,103	13,017 (92.3)	9819 (69.6)	3523 (25.0)	10,089 (71.5)
*Tricholoma terreum*	MG45	424	20	15,815	17,962	16,569 (92.2)	12,503 (69.6)	4286 (23.9)	11,972 (66.7)
*Praearthromyces corneri*	D33	524	0	15,846	18,802	18,339 (97.5)	12,342 (65.6)	4662 (24.8)	12,961 (78.9)
*Rugosomyces carneus*	D47	340	0	16,549	18,640	16,597 (89.0)	13,393 (71.9)	5428 (29.1)	13,092 (70.2)
*Cystodermella granulosa*	DBG-29008	248	6	17,651	20,236	18,717 (92.5)	15,841 (78.3)	6974 (34.5)	15,095 (74.6)
*Flagelloscypha* sp.	PMI 526	410	14	26,599	31,410	31,146 (99.2)	18,012 (57.3)	4205 (13.4)	22,038 (70.2)
*Macrocystidia cucumis*	KM177596	- *	20	11,806	13,180	12,344 (93.7)	10,079 (76.5)	3441 (26.1)	10,732 (81.4)

* The genome is too fragmented to allow for reliable tRNA prediction.

**Table 3 jof-10-00746-t003:** The orthogroup analysis result.

Total number of species/strains	19
Total number of proteins	325,167
Total number of orthogroups	34,538
Number of orthogroups with homologous proteins	22,708
Number of orthogroups with single protein (unassigned proteins)	11,830
Number of orthogroups with all species present	4131
Number of single-copy orthogroups	204

## Data Availability

The genome sequences, repeat and gene annotation, coding and protein sequences, and functional annotation data were deposited to FigShare and available at https://doi.org/10.6084/m9.figshare.26869750.v1.

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
