# Peer review of "Comparative Genomics Reveals Species-Specific Genes and Symbiotic Adaptations in Tricholoma matsutake"

_jof, 2024, doi:10.3390/jof10110746_

Round 1

Reviewer 1 Report

This paper attempts to clarify the biological characteristics of matsutake mushrooms based on comparative genomic analysis of Tricholoma and other taxa that are taxonomically related to matsutake mushrooms. As the authors state, many of the discussions based on the basic structure of the genome or the presence or absence of predicted genes are novel and very interesting. However, these are only hypotheses based on genome data, and without actual functional analysis (such as experiments using hyphal cells, or detection of enzymes and compounds), it is impossible to confirm whether they are true or not. Many of the discussions are abstract, and it seems that they are being forced to link with known findings.

Also, in comparative genomics, while the comparison of the section Matsutake and other sections of the genus Tricholoma is good, the comparison with the family Tricholomataceae other than the genus Tricholoma needs to be significantly revised. The Tricholomataceae used by the authors is not objective, as they use different meanings for different contexts. This is because, according to recent molecular phylogenetic data, all species other than those in the genus Tricholoma analyzed by the authors are not included in the Tricholomataceae, and the taxonomic rank that can include them is the order Agaricales. It remains a question as to why the authors did not use taxa other than the genus Tricholoma that belong to the true Tricholomataceae. By addressing these points, I think it would be possible to explain the genome evolution and characteristics of Matsutake to the readers of this paper.

P2, Table 1: Materials

As indicated in the main text, it is clear that Tricholoma furcatifolium is not a member of the genus Tricholoma, so a name such as Praearthromyces sp. should have been used from the outset. Rugosomyces carneus is in the family Lyophyllaceae,Cystodermella granulosa is in the family AgaricaceaeFlagelloscypha sp. is in the Niacece, and Macrocystidia cucumisis in the family Macrocystidiaceae. In the Lyophyllaceae, which are closely related to the true Tricholomataceae, the genome data for Lyophyllum shimeji has been made public, and so this should be used from the perspective of comparative genomics.

P2, L53: host

T. matsutake is ectomycorrhizal with PinusAbiesTsuga, and Picea in the family Pinaceae at least. If the populations in western China are considered to be T. matsutakeQuercus and Castanopsis can also be hosts. For details, see Aoki et al. (2022) Mycoscience 63: 197–214.

P3, L69: “Tricholomataceae”

The authors should clarify what “Tricholomataceae” means from a taxonomic (molecular phylogenetic) perspective. As mentioned in relation to Table 1, the authors' definition of “Tricholomataceae” is ambiguous and not correct from a taxonomic perspective. If anything, it would be Agaricales.

P4, L128, L130:

As mentioned in Table 2, T. furcatifolium is a saprotrophic fungus that belongs to the genus Praearthromyces (family Lyophyllaceae), so the analysis and discussion should be based on this premise.

P12, L286-287:

Matsutake mushrooms grow in iron-poor soil. The authors should recheck the references cited (especially Yamanaka et al. 2020, which summarizes a large number of Japanese research data).

P14, L351-355

This kind of argument is abstract and lacks concreteness, and cannot be considered constructive.

P14, L367-370

This argument gives the impression that matsutake mushrooms use lignocellulose as a carbon source, but is this really the case? If so, it is inconsistent with the fact that matsutake mushroom mycelium growth is slow and that monosaccharides are used in ordinary culture media (the level of disaccharide utilization is also low).

P14, L376-377

This argument also fails to guarantee scientific validity because it is based on a simple experiment using a single strain of T. matsutake (in Japan, there are many known cultivation studies in T. matsutake).

Author Response

Reviewer 1

Major comments

This paper attempts to clarify the biological characteristics of matsutake mushrooms based on comparative genomic analysis of Tricholoma and other taxa that are taxonomically related to matsutake mushrooms. As the authors state, many of the discussions based on the basic structure of the genome or the presence or absence of predicted genes are novel and very interesting. However, these are only hypotheses based on genome data, and without actual functional analysis (such as experiments using hyphal cells, or detection of enzymes and compounds), it is impossible to confirm whether they are true or not. Many of the discussions are abstract, and it seems that they are being forced to link with known findings.

Thank you very much for your interest in our study. As per the reviewer’s suggestion, we have strengthened the Discussion section with more detailed content. In the Detailed Comment section, we explain the revisions by indicating the corresponding page and line numbers. Additionally, we conducted further PCR amplification experiments to partially validate the findings from our comparative genomics analysis (Supplementary Figure 7).

Also, in comparative genomics, while the comparison of the section Matsutake and other sections of the genus Tricholoma is good, the comparison with the family Tricholomataceae other than the genus Tricholoma needs to be significantly revised. The Tricholomataceae used by the authors is not objective, as they use different meanings for different contexts. This is because, according to recent molecular phylogenetic data, all species other than those in the genus Tricholoma analyzed by the authors are not included in the Tricholomataceae, and the taxonomic rank that can include them is the order Agaricales. It remains a question as to why the authors did not use taxa other than the genus Tricholoma that belong to the true Tricholomataceae. By addressing these points, I think it would be possible to explain the genome evolution and characteristics of Matsutake to the readers of this paper.

To analyze Tricholoma-specific genes, we used all assemblies found at the time through an NCBI search for “Tricholomataceae” (Taxonomy ID: 5351). As the reviewer pointed out, from a taxonomic perspective, this may not represent true Tricholomataceae. Therefore, we have excluded it from the title and replaced "Tricholomataceae" with Tricholoma and outgroup species, or the suggested Agaricales.

Detail comments

P2, Table 1: Materials

As indicated in the main text, it is clear that Tricholoma furcatifolium is not a member of the genus Tricholoma, so a name such as Praearthromyces sp. should have been used from the outset. Rugosomyces carneus is in the family Lyophyllaceae, Cystodermella granulosa is in the family Agaricaceae, Flagelloscypha sp. is in the Niacece, and Macrocystidia cucumisis in the family Macrocystidiaceae. In the Lyophyllaceae, which are closely related to the true Tricholomataceae, the genome data for Lyophyllum shimeji has been made public, and so this should be used from the perspective of comparative genomics.

As the reviewer suggests, we have replaced Tricholoma furcatifolium with Praearthromyces. Additionally, following previous research indicating that T. furcatifolium is Praearthromyces (van de Peppel et al. 2022), we downloaded all available Praearthromyces nucleotide sequences from NCBI and performed a BLAST search on Tricholoma furcatifolium, confirming that it is Praearthromyces corneri. This information has been added to the main text (P5 L147) and Supplementary Figure 3.

Thank you again for pointing out that the Tricholomataceae species obtained from the NCBI search may not represent true Tricholomataceae. We have adjusted the comparison in the paper to focus on Tricholoma and outgroup species. Although adding Lyophyllum shimeji would have been ideal, it was not included because it did not appear in the Tricholomataceae search results on NCBI. We appreciate your understanding on this matter.

P2, L53: host

T. matsutake is ectomycorrhizal with Pinus, Abies, Tsuga, and Picea in the family Pinaceae at least. If the populations in western China are considered to be T. matsutake, Quercus and Castanopsis can also be hosts. For details, see Aoki et al. (2022) Mycoscience 63: 197–214.

Thank you for providing more information about the host. We have added the hosts of Tricholoma matsutake other than Pinus and the reference you suggested to the Introduction section (P3 L46).

P3, L69: “Tricholomataceae”

The authors should clarify what “Tricholomataceae” means from a taxonomic (molecular phylogenetic) perspective. As mentioned in relation to Table 1, the authors' definition of “Tricholomataceae” is ambiguous and not correct from a taxonomic perspective. If anything, it would be Agaricales.

As the reviewer’s suggestion, we have minimized the use of "Tricholomataceae" throughout the manuscript. In the specified line, we retained "Tricholomataceae" only in the Methods section to explain the NCBI search method, adding the taxonomy ID for clarity.

P4, L128, L130:

As mentioned in Table 2, T. furcatifolium is a saprotrophic fungus that belongs to the genus Praearthromyces (family Lyophyllaceae), so the analysis and discussion should be based on this premise.

As mentioned earlier, we identified T. furcatifolium as Praearthromyces corneri, and this information has been added to the main text (P5 L147) and Supplementary Figure 3.

P12, L286-287:

Matsutake mushrooms grow in iron-poor soil. The authors should recheck the references cited (especially Yamanaka et al. 2020, which summarizes a large number of Japanese research data).

Yamanaka et al. (2020) explains that matsutake mushrooms grow in nutrient-poor soil and also mentions that iron ions (Fe3+) and iron citrate are beneficial for their growth. The term "nutrient" in this paper seems to refer to carbon sources rather than minerals. We have added additional references indicating that the soil in matsutake habitats is rich in iron. Furthermore, we have revised the text to state that matsutake grows in nutrient-poor but generally iron-rich environments (P16 L457).

P14, L351-355

This kind of argument is abstract and lacks concreteness, and cannot be considered constructive.

We have revised and strengthened the content of the discussion section (P15 L396).

P14, L367-370

This argument gives the impression that matsutake mushrooms use lignocellulose as a carbon source, but is this really the case? If so, it is inconsistent with the fact that matsutake mushroom mycelium growth is slow and that monosaccharides are used in ordinary culture media (the level of disaccharide utilization is also low).

As the reviewer suggested, we made an error in our previous description. The GO terms associated with the section Matsutake are related to interactions with the external environment, specifically involving chitin, not lignocellulose. We have revised our discussion to reflect the correct GO terms associated with the section Matsutake (P15 L423-432).

P14, L376-377

This argument also fails to guarantee scientific validity because it is based on a simple experiment using a single strain of T. matsutake (in Japan, there are many known cultivation studies in T. matsutake).

We have strengthened the discussion by adding more references (P16 L439).

Reviewer 2 Report

Major Comments:

Figure 5 – Clarity and Detail: Figure 5 is visually confusing and lacks clear focus on the most relevant pathways. I suggest selecting and amplifying the key pathways to improve clarity and align the figure more closely with the main findings discussed in the text. Additionally, further detailed discussion of these pathways within the manuscript would enhance understanding.

Pathway Discussion: The manuscript would benefit from a deeper exploration of certain pathways, such as those related to iron ion homeostasis and tryptophan metabolism. A more thorough analysis of how these gene losses and gains impact T. matsutake's adaptation to its environment could significantly strengthen the discussion.

Visual Representation: Overall, clearer labeling and more detailed legends in figures would improve the readability and interpretation of the genomic data presented. This will ensure that complex information is accessible and easier to follow.

The manuscript provides a comprehensive and insightful comparative genomic analysis of the Tricholomataceae family, particularly on Tricholoma matsutake, an ectomycorrhizal fungus of significant commercial value. The well-structured study contributes valuable information to our understanding of the evolutionary mechanisms governing symbiotic relationships between these fungi and their host trees. The work highlights several intriguing genomic features, including the prominence of repetitive elements and retrotransposons and the expansion and contraction of gene families related to important metabolic processes.

Good points:

Relevance and Originality: The focus on the genomic evolution of T. matsutake provides critical insights into the adaptations required for its host-dependent lifestyle, which could have significant implications for its cultivation and commercial potential. The analysis of repetitive elements and retrotransposons, particularly the dominance of LTR Gypsy elements, is novel and sheds light on genome architecture.

Phylogenomic Analysis: The phylogenomic positioning of T. matsutake within the Tricholomataceae family, alongside related species such as T. bakamatsutake, is robust and provides clarity on the evolutionary history of these fungi. This reinforces the monophyletic grouping and evolutionary pathways the manuscript seeks to address.

Gene Family Analysis: The discussion of gene family expansion and contraction, particularly the focus on tryptophan metabolism and iron ion homeostasis, offers deep insights into the adaptive mechanisms of T. matsutake in nutrient-poor environments. This finding is crucial for understanding the biological and ecological success of this species.

Points for Improvement:

Figure 5 – Lack of Clarity: One of the major shortcomings of the manuscript lies in Figure 5, which presents critical pathways related to the genomic adaptations of T. matsutake. The image is visually confusing and cluttered, making it difficult for the reader to extract the most important information. To improve this, I suggest selecting and amplifying the most relevant pathways that are directly discussed in the text. By focusing on fewer, key pathways, the figure will be easier to interpret and align better with the manuscript's narrative. Furthermore, more detailed discussion and interpretation of these pathways in the text would enhance the clarity and impact of the findings.

Discussion of Specific Pathways: While the genomic data is comprehensive, certain key pathways, such as those involved in iron ion homeostasis and tryptophan metabolism, could benefit from deeper exploration. The authors could elaborate further on how the loss and gain of specific genes in these pathways contribute to the ecological fitness and symbiotic relationships of T. matsutake in more detail.

Visual Representation of Genomic Data: Beyond Figure 5, other figures would benefit from clearer labeling and more detailed legends to guide readers through complex genomic data. Visual representation is crucial in a study of this nature, and improving the clarity of these elements would significantly enhance the overall presentation. The authors should improve the legends to give a more auto-informative context.

In conclusion, this manuscript is valuable in studying ectomycorrhizal fungi, specifically T. matsutake, and its genomic adaptations. However, the manuscript would benefit from revisions to improve the clarity of its figures, particularly Figure 5, and to deepen the discussion of certain key pathways. Once these adjustments are made, the work has the potential to be a highly impactful publication in the field of fungal genomics and evolutionary biology.

Author Response

Reviewer 2

Major Comments:

Figure 5 – Clarity and Detail: Figure 5 is visually confusing and lacks clear focus on the most relevant pathways. I suggest selecting and amplifying the key pathways to improve clarity and align the figure more closely with the main findings discussed in the text. Additionally, further detailed discussion of these pathways within the manuscript would enhance understanding.

Thank the reviewer for the advice. We extracted only the key pathways as the reviewer suggested for Figure 5 (now Figure 6), and additionally, we included conservation data for the ortholog alignment between T. matsutake and T. bakamatsutake. The original figure was moved to Supplementary Figure 6, and we have added this information to the discussion section.

Pathway Discussion: The manuscript would benefit from a deeper exploration of certain pathways, such as those related to iron ion homeostasis and tryptophan metabolism. A more thorough analysis of how these gene losses and gains impact T. matsutake's adaptation to its environment could significantly strengthen the discussion.

We have added these details more thoroughly to the discussion section (P16 L451-476).

Visual Representation: Overall, clearer labeling and more detailed legends in figures would improve the readability and interpretation of the genomic data presented. This will ensure that complex information is accessible and easier to follow.

As requested, we have improved the quality of the figures.

Detail comments

The manuscript provides a comprehensive and insightful comparative genomic analysis of the Tricholomataceae family, particularly on Tricholoma matsutake, an ectomycorrhizal fungus of significant commercial value. The well-structured study contributes valuable information to our understanding of the evolutionary mechanisms governing symbiotic relationships between these fungi and their host trees. The work highlights several intriguing genomic features, including the prominence of repetitive elements and retrotransposons and the expansion and contraction of gene families related to important metabolic processes.

Good points:

Relevance and Originality: The focus on the genomic evolution of T. matsutake provides critical insights into the adaptations required for its host-dependent lifestyle, which could have significant implications for its cultivation and commercial potential. The analysis of repetitive elements and retrotransposons, particularly the dominance of LTR Gypsy elements, is novel and sheds light on genome architecture.

Phylogenomic Analysis: The phylogenomic positioning of T. matsutake within the Tricholomataceae family, alongside related species such as T. bakamatsutake, is robust and provides clarity on the evolutionary history of these fungi. This reinforces the monophyletic grouping and evolutionary pathways the manuscript seeks to address.

Gene Family Analysis: The discussion of gene family expansion and contraction, particularly the focus on tryptophan metabolism and iron ion homeostasis, offers deep insights into the adaptive mechanisms of T. matsutake in nutrient-poor environments. This finding is crucial for understanding the biological and ecological success of this species.

Points for Improvement:

Figure 5 – Lack of Clarity: One of the major shortcomings of the manuscript lies in Figure 5, which presents critical pathways related to the genomic adaptations of T. matsutake. The image is visually confusing and cluttered, making it difficult for the reader to extract the most important information. To improve this, I suggest selecting and amplifying the most relevant pathways that are directly discussed in the text. By focusing on fewer, key pathways, the figure will be easier to interpret and align better with the manuscript's narrative. Furthermore, more detailed discussion and interpretation of these pathways in the text would enhance the clarity and impact of the findings.

We have made the revisions as requested by the reviewer and added a figure related to iron-associated genes. The original Figure 5 has now been updated as Figure 6.

Discussion of Specific Pathways: While the genomic data is comprehensive, certain key pathways, such as those involved in iron ion homeostasis and tryptophan metabolism, could benefit from deeper exploration. The authors could elaborate further on how the loss and gain of specific genes in these pathways contribute to the ecological fitness and symbiotic relationships of T. matsutake in more detail.

We have added this discussion in detail to the discussion section (P16 L451-476).

Visual Representation of Genomic Data: Beyond Figure 5, other figures would benefit from clearer labeling and more detailed legends to guide readers through complex genomic data. Visual representation is crucial in a study of this nature, and improving the clarity of these elements would significantly enhance the overall presentation. The authors should improve the legends to give a more auto-informative context.

We have made the revisions as per the reviewers’ suggestions.

In conclusion, this manuscript is valuable in studying ectomycorrhizal fungi, specifically T. matsutake, and its genomic adaptations. However, the manuscript would benefit from revisions to improve the clarity of its figures, particularly Figure 5, and to deepen the discussion of certain key pathways. Once these adjustments are made, the work has the potential to be a highly impactful publication in the field of fungal genomics and evolutionary biology.

Reviewer 3 Report

Tricholoma, as the highly valued ectomycorrhizal fungus, always is of interest studying focus. This manuscript presents a comprehensive comparative genomic analysis of species within the Tricholomataceae family. Genomic data from 19 assemblies representing 13 species were analyzed to identify. The results provided the conclusion that T. matsutake forms a monophyletic group closely related to T. bakamatsutake. Gene family expansion and contraction analyses highlighted the unique evolutionary pressures on T. matsutake, particularly the loss of tryptophan-related metabolic pathways and the gain of genes related to iron ion homeostasis genus-, species-, and strain-specific genes, revealing significant evolutionary adaptations. The results help readers to understand genus Tricholoma better. However, I have two questions:  

1. I noticed that author used multiple publicly available genomic datasets from NCBI, and it seems that the data published by Professor Li and H does not include annotated files. If author only use the de novo annotation strategy and do not include evidence such as transcriptomics, it is inevitable that too many pseudogenes will be introduced. Because the main analysis focuses on the expansion and contraction of gene families, this may lead to biased results. At the same time, the integrity of the BUSCO genome of Cystodermella granulosa that author using is not very good. It belongs to the honey fungus, and perhaps you can discard it. The genomic data of Tricholoma furcatifolium may not be reliable. I did not find its species classification in the ITS of the genome, and from the phylogenetic tree, author can also see that it is too far away from the genus Tricholoma. This may be incorrect data. I hope author can identify its species or remove it from the results.

2. In the results, GO annotation is used to illustrate the point of view, but I think that more genes may not be obtained through GO annotation. Of course, author also made a bar chart to illustrate this situation. Therefore, the explanation of GO functional enrichment results should be more cautious. I hope author can explain it in the discussion.

The manuscript is good at details.

Author Response

Reviewer 3

Major comments

Tricholoma, as the highly valued ectomycorrhizal fungus, always is of interest studying focus. This manuscript presents a comprehensive comparative genomic analysis of species within the Tricholomataceae family. Genomic data from 19 assemblies representing 13 species were analyzed to identify. The results provided the conclusion that T. matsutake forms a monophyletic group closely related to T. bakamatsutake. Gene family expansion and contraction analyses highlighted the unique evolutionary pressures on T. matsutake, particularly the loss of tryptophan-related metabolic pathways and the gain of genes related to iron ion homeostasis genus-, species-, and strain-specific genes, revealing significant evolutionary adaptations. The results help readers to understand genus Tricholoma better. However, I have two questions: 

  1. I noticed that author used multiple publicly available genomic datasets from NCBI, and it seems that the data published by Professor Li and H does not include annotated files. If author only use the de novo annotation strategy and do not include evidence such as transcriptomics, it is inevitable that too many pseudogenes will be introduced. Because the main analysis focuses on the expansion and contraction of gene families, this may lead to biased results.

Thank you for your suggestion regarding potential issues with gene family expansion and contraction analysis. We are also aware that some publicly available annotation data for T. matsutake is missing. To address this, we performed evidence-based structural annotation using not only de novo prediction but also 330,798 Tricholomatineae (a suborder of basidiomycete fungi in the order Agaricales) proteins downloaded from NCBI (P3, Line 82). This dataset includes 23,622 T. matsutake proteins, which we believe can sufficiently serve as a substitute for transcriptome data.

At the same time, the integrity of the BUSCO genome of Cystodermella granulosa that author using is not very good. It belongs to the honey fungus, and perhaps you can discard it. The genomic data of Tricholoma furcatifolium may not be reliable. I did not find its species classification in the ITS of the genome, and from the phylogenetic tree, author can also see that it is too far away from the genus Tricholoma. This may be incorrect data. I hope author can identify its species or remove it from the results.

Although the BUSCO analysis results for Cystodermella granulosa were not favorable, we believe it is necessary to use this species as an outgroup in order to eliminate as many shared genes as possible between the outgroup and Tricholoma, facilitating a more precise analysis of the genomic characteristics of the Tricholoma genus.

For T. furcatifolium, we downloaded all nucleotide sequences related to Praearthromyces from NCBI and mapped them to the T. furcatifolium genome using BLAST. The results showed that the Translation Elongation Factor 1A (OM974133.1) of Praearthromyces corneri had 100% query coverage and 100% identity, indicating that T. furcatifolium is actually Praearthromyces corneri. This information has been added to the manuscript and supplementary Figure 3, and T. furcatifolium has been revised to P. corneri.

  1. In the results, GO annotation is used to illustrate the point of view, but I think that more genes may not be obtained through GO annotation. Of course, author also made a bar chart to illustrate this situation. Therefore, the explanation of GO functional enrichment results should be more cautious. I hope author can explain it in the discussion.

Previously, we mentioned in Results that only about 65.6% of the proteins in T. matsutake were assigned GO terms (P7, L205). While the functions of the remaining proteins are mostly unknown, we believe that the significant functions discovered through enrichment tests in core orthogroups and orthogroups that underwent expansion or contraction help explain the biological characteristics of T. matsutake and its related species at the genomic level. However, in the discussion, we added that the incomplete annotation limits a comprehensive understanding of the full functions of these orthogroups as the reviewer suggested (P15 L413-422).